# Resolving Complex Social Dilemmas by Aligning Preferences with Counterfactual Regret

## Abstract

Social dilemmas are situations where gains from cooperation are possible but misaligned incentives make it hard to find and stabilize prosocial joint behavior. In such situations selfish behaviors may harm the social good. In spatiotemporally complex social dilemmas, the barriers to cooperation that emerge from misaligned incentives interact with obstacles that stem from spatiotemporal complexity. In this paper, we propose a multi-agent reinforcement learning algorithm which aims to find cooperative resolutions for such complex social dilemmas. Agents maximize their own interests while also helping others, regardless of the actions their co-players take. This approach disentangles the causes of selfish reward from the causes of prosocial reward. Empirically, our method outperforms multiple baseline methods in several complex social dilemma environments.

## 1 Introduction

Individuals often have their own desires which do not align with their group's objectives. This kind of misalignment is common in practical situations. For example, in economic cooperation, participants could gain by investing in trust and rule compliance to ultimately enhance market efficiency and growth, but must avoid the temptation to chase short-term gains by deceit or rule evasion. When this kind of scenario also contains spatial and temporal complexity it is called a Sequential Social Dilemmas (SSD) Leibo et al. (2017). The reason SSDs pose challenging environments for learning agents is that their spatial and temporal complexity can interact with the strategic complexity arising from the agents' misaligned incentives.

For example, in one classic SSD game, called Cleanup (Hughes et al., 2018), players are rewarded by collecting apples from an orchard whose growth is restricted by the accumulation of pollution in a nearby river. Apples stop growing unless players contribute to the public good by cleaning the river, a task which involves navigating to a specific location and executing multi-step action sequences. The cleaner the river, the faster the apples grow. However, since the river and orchard are geographically separated, players cannot eat and clean at the same time, and must also spend time walking between the two locations. Selfish players who never clean benefit more from a clean river than do altruistic players who perform all the work of cleaning, since the altruists lose time cleaning and walking between the river and orchard. Cleaning the river promotes higher long-term collective return (it is prosocial), but it requires necessary sacrifice on the part of the individuals who spend their time cleaning rather than eating. The cooperation between "cleaners" and "eaters" in Cleanup is an example of division of labor where some roles are remunerated less than other roles, a common though unfair arrangement in real life (Yaman et al., 2023).

Owing to their real-world significance, sequential social dilemmas have recently attracted much attention from researchers. Many works attempt to promote cooperation behaviors by learning relationships between agents' actions. Algorithms such as LOLA (Foerster et al., 2017) promote cooperation by modeling opponents' behaviors. Jaques et al. (2019) investigate the causal relationships between the actions of agents. Modeling the relationships between actions might lead to reciprocation behaviors, as such methods could end up capturing spurious correlations between behaviors while failing to identify real causal relationships between actions and outcomes. In another line of work (Hughes et al., 2018; McKee et al., 2020; Wang et al., 2019; Lupu & Precup, 2020; Kwon et al., 2023), agents are encouraged to intrinsically maximize the welfare of others to promote cooperation behavior among the group. The Gifting mechanism (Lupu & Precup, 2020) allocates a portion of

agents' individual rewards to their co-players, encouraging collaboration by decreasing the cost of self-sacrifice; LIO (Yang et al., 2020) promotes cooperative paradigms by learning to incentivize other agents using agent's own rewards. Furthermore, there are also methods like (Kwon et al., 2023) that attempt to automatically align agents' incentives with global incentives. These approaches design altruistic rewards to promote cooperation behaviors, but fail to capture the reward generation process, which may lead to spurious prediction of the true team incentives.

In this work, we aim to establish a reinforcement learning algorithm using counterfactual regret to align incentives in a group of agents through maximizing both the agent's individual outcome and other agents' outcomes. In SSDs, naively using individual for each agent might not always align with the group's objective. Because the agents might be rewarded for some selfish behaviors when other agents cooperate (e.g. by exploiting them). Such entanglement of the agents' policies would bring bias to the estimation of their contributions to the society. Furthermore, such entanglement may cause spurious prediction on the real cause of the agents' reward. Observing that, we utilize a causal model to mimic the generation process of the individual rewards for all other agents in the group. Counterfactual regret has been used to solve problems under single agent's scenarios (Brown et al., 2019). Based on that model, we could define the counterfactual regret in multi-agent setting as the difference between the maximum counterfactual reward for other agents and the other agents' actual reward. More specifically, we calculate the maximum expected outcome for all other agents by predicting their maximum expected rewards under multiple counterfactual scenarios, then subtract the current other agents' reward to construct counterfactual regret. Minimizing such counterfactual regret could guide each individual agent to additionally consider other agents' reward, which would eventually lead to better cooperation paradigm. As the basis of method, we utilize a causal model to describe the generative process within the Partially Observable Markov Games and guide the counterfactual reasoning. Assisted by such a causal model, we aim to capture the real cause of reward generation, reducing the risk of learning spurious relationships and generating counterfactual rewards by intervening on the actions of agents. Theoretically, we prove that under faithfulness assumption and Markov condition, we can identify the real cause in the generation of individual rewards, which enable us to reason the counterfactual rewards of agents. Furthermore, we demonstrate that our method surpass the baseline methods in most of the sequential social dilemma environments through empirical results.

In summary, our contributions are threefold. Firstly, we exploit a generative model to explicitly capture the generation process of individual rewards in SSDs. It provides a guidance for the further counterfactual reasoning from the causal view. Secondly, we infer the counterfactual regret based on our learned causal model to mimic the expected outcome of other agents. Combining this counterfactual regret with the original individual rewards guides our agents to learn and respond to the social incentives embedded in their environment. Lastly, we evaluate our algorithm on four sequential social dilemma tasks, along with their respective variants, to assess its performance comprehensively. The experimental results demonstrate the superior performance of our algorithm in fostering cooperation and enhancing overall collective reward.

## 2 RELATED WORK

Below we review the related work on intrinsic reward design methods and causality-facilitated reinforcement learning methods.

In the SSD setting, intrinsic motivation methods allow agents to actively care about the welfare of others intrinsically, or modify the extrinsic rewards of other agents. In scenarios where environmental rewards are misleading, relying solely on external rewards provided by the environment may not be sufficient for effective learning. Social learning is incredibly important for humans and has been linked to our ability to achieve unprecedented progress and coordination on a massive scale (Henrich, 2015; Harari, 2014; Laland, 2017; Van Schaik & Burkart, 2011; Herrmann et al., 2007). While some previous work has investigated intrinsic social motivation for reinforcement learning under sequential social dilemma setting, $e.g.$, Sequeira et al. (2011); Hughes et al. (2018); Peysakhovich & Lerer (2017), these approaches rely on hand-crafted rewards specific to the environment, or allowing agents to view the rewards obtained by other agents (Durugkar et al., 2020). Methods like D3C (Gemp et al., 2020) and Auto-aligning multi-agent incentives (Kwon et al., 2023) take further steps to align agents' incentives automatically by modifying agents' incentives online to achieve a new goal. This

could help decentralized agents automatically modify their incentives based on the preset incentives online to achieve a new goal. However minimizing the Price of anarchy directly results in increased inequality (Gemici et al., 2018). Achieving coordination among agents in sequential social dilemmas still remains a difficult problem. Prior work in this domain, *e.g.*, Foerster et al. (2016; 2018), often resorts to centralized training to ensure that agents learn to coordinate. While communication among agents could help with coordination, training emergent communication protocols also remains a challenging problem; recent empirical results underscore the difficulty of learning meaningful emergent communication protocols, even when relying on centralized training *e.g.* Cao et al. (2018); Forestier & Oudeyer (2017).

Causality is used to address many RL problems like improving the transfer ability of RL agents(Huang et al., 2021; Feng et al., 2022), model-based RL Zhang & Bareinboim (2016); Liu et al. (2023). (Hu et al., 2023a; Pitis et al., 2022; Hu et al., 2023b) unveil the causal structures within the MDP generative process and exploit those causal lenses to facilitate policy learning. More recently, causality-inspired methods are proposed to address MARL problems (Grimbly et al., 2021; Jaques et al., 2019; Li et al., 2021). Li et al. (2021) introduce counterfactual Shapley value in the credit assignment setting. (Zhou et al., 2022) attempted to use counterfactual prediction in the value decomposition field. Despite the significant contributions of prior research, a common oversight has been the neglect of self-interest settings, where individual interests may not necessarily align with team goals. Even in studies that acknowledge this aspect, there's often an incomplete capture of each individual's incentives, potentially leading the algorithms to converge to suboptimal solutions. In contrast, our work delves into the causal mechanisms underlying the generation of agents' individual rewards, facilitating a more effective alignment of individual interests with the collective objectives. Moreover, by employing counterfactual reasoning, we mitigate the influence of other agents' actions, leading to a more stable and precise estimation of the overall incentives.

## 3 PRELIMINARY

**Partially Observable Markov Game (POMG)** is defined by the tuple $\langle N, S, O, T, A, R \rangle$, in which multiple agents are trained to independently maximize their own individual reward; The environment state is given by $s \in S$. At each timestep $t$, each agent $i \in N$ chooses an action $a_t^i \in A$. The actions of all $N$ agents are combined to form a joint action $\boldsymbol{a}_t = [a_t^1, \cdots, a_t^N]$, which produces a transition in the environment $T(s_{t+1}|\boldsymbol{a}_t, s_t)$, according to the state transition $T$. We consider a partially observable setting in which the $i$-th agent can only view a portion of the true state, represented as individual observation $o_t^i$. We denote all agents' observation as the joint observation $\boldsymbol{o}_t$. Each agent $i$ seeks to maximize its own total expected discounted future reward, $R^i = \sum_{t=0}^{\infty} \gamma^t r_t^i$, where $\gamma$ is the discount factor. Each agent $i$ then receives its own reward $r^i(\boldsymbol{a}_t, s_t)$, which depends on the actions of other agents.

**Counterfactual Reasoning** in our paper refers to reasoning that all individual rewards $\boldsymbol{r}$ would be $\boldsymbol{r}^{\text{cf}}$ if the collective actions $\boldsymbol{a}$ had been $\boldsymbol{a}^{\text{cf}}$ at the latent state $\boldsymbol{s} = \boldsymbol{s}$. We exploit the learned causal model to estimate the latent state variable $\boldsymbol{s}$ and infer the counterfactual reward $\boldsymbol{r}^{\text{cf}}$ given counterfactual action $\boldsymbol{a}^{\text{cf}}$. To interpret the phrase: had collective actions $\boldsymbol{a}$ been $\boldsymbol{a}^{\text{cf}}$, modify the original model and replace the equation for $\boldsymbol{a}$ by a constant $\boldsymbol{a}^{\text{cf}}$. This replacement permits the constant $\boldsymbol{a}^{\text{cf}}$ to differ from the actual value of $\boldsymbol{a}$ without rendering the system of equations inconsistent (Pearl, 2010). In general, it can be shown (Pearl (2009), Section 3) that, whenever the graph is Markovian (i.e., acyclic with independent exogenous variables) the post-interventional distribution $P(\boldsymbol{r} = \boldsymbol{r}^{\text{cf}}|do(\boldsymbol{a} = \boldsymbol{a}^{\text{cf}}))$ is given by the following expression $P(\boldsymbol{r} = \boldsymbol{r}^{\text{cf}}|do(\boldsymbol{a} = \boldsymbol{a}^{\text{cf}}), \boldsymbol{s}) = P(\boldsymbol{r}^{\text{cf}} \mid \boldsymbol{a}^{\text{cf}}, \boldsymbol{s})P(\boldsymbol{s})$. If there is no incoming path for $\boldsymbol{a}$ in the causal graph, i.e., $\boldsymbol{a}$ has no causal parents, we have $P(\boldsymbol{r} = \boldsymbol{r}^{\text{cf}}|do(\boldsymbol{a} = \boldsymbol{a}^{\text{cf}}), \boldsymbol{s}) = P(\boldsymbol{r} \mid \boldsymbol{a}^{\text{cf}}, \boldsymbol{s})$.

## 4 METHODOLOGY

In this paper, we address the challenge of optimizing collective reward within the framework of sequential social dilemma (SSD), which strikes a balance between the pursuit of individual rewards and the achievement of communal benefits. Our objective is to facilitate the policy learning process of agents to align the agents' preferences with the collective outcome. To this end, we establish intrinsic reward for the agents to care more about others' welfare while maximizing their individual

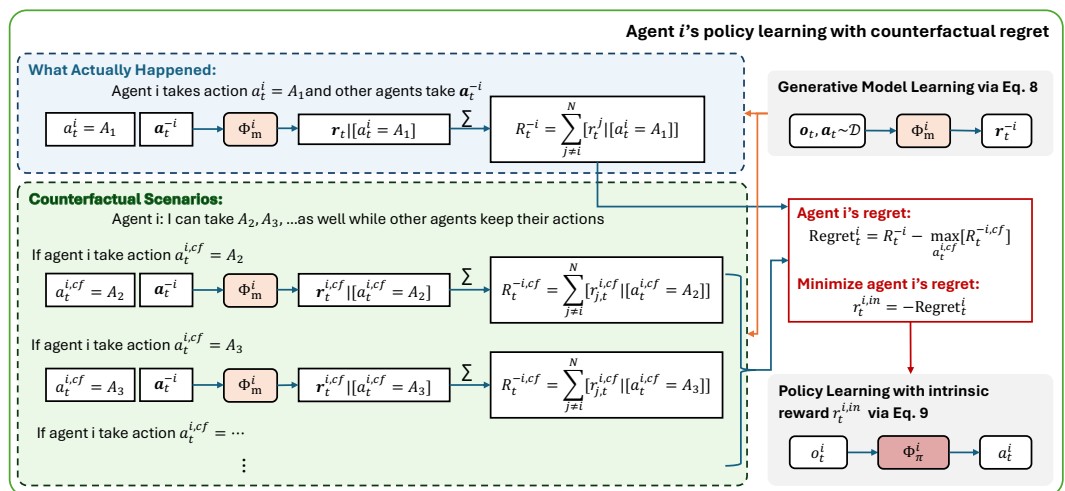

Figure 1: The figure pictures the overall training and inference process of a single agent $i$. Blue blocks represent the generated process of the actual reward; green blocks denote the counterfactual reward generation process; red blocks represent the regret calculation process and intrinsic reward construction process; gray blocks represent two learning process with model parameters $\Phi_m$ and $\Phi_\pi$.

interests. We define the counterfactual regret in multi-agent setting as the difference between the agents' optimal prosocial behaviors and its current action. The intrinsic reward could be defined as the negative of counterfactual regret. Therefore, we could promote agents' cooperative behaviors by maximizing the intrinsic reward.

Figure 1 depicts the framwork of agent $i$'s policy learning process. The joint action $\boldsymbol{a}_t$ is sampled from the training data. We intervene agent $i$'s action to get counterfactual actions(e.g, $A_2, A_3$). The counterfactual action $a_t^{i,\mathrm{cf}}$ are input to the causal model $\Phi_m^i$ along with the joint observation vectors $\boldsymbol{o}_t$ and other agents' actions $\boldsymbol{a}_t^{-i}$. The output counterfactual rewards $\boldsymbol{r}_t^{-i,\mathrm{cf}}$ are summed up to generate the collective counterfactual reward $R_t^{-i,\mathrm{cf}}$. Such collective counterfactual reward are used to calculate the counterfactual regret along with actual collective counterfactual reward $R_t^{-i}$. In order to minimize such counterfactual regret, we construct the intrinsic reward $r_t^{i,\mathrm{in}}$ as $-\mathrm{Regret}_t^i$. In the policy learning process, we combine such intrinsic reward $\boldsymbol{r}_t^{\mathrm{in}}$ with extrinsic reward $\boldsymbol{r}_t^{\mathrm{ex}}$ to assist our agents' policy learning.

## 4.1 OVERVIEW

In SSD, each agent act independently in the environment. Therefore, we take agent $i$ as an example to illustrate our method. Each agent $i$ consists of a generative model $\Phi_m^i$ and a policy model $\Phi_\pi^i$. **Generative model** $\Phi_m^i$ parameterizes the generation of individual rewards in POMG given the joint observation $\boldsymbol{o}_t$ and joint action $\boldsymbol{a}_t$. **Policy model** $\Phi_\pi^i$ takes agent $i$'s individual observation $o_t^i$ as input and output its individual action $a_t^i$. We define the overall objective function for agent $i$ as:

$$L^i(\Phi_m^i, \Phi_\pi^i) = L_m^i(\Phi_m^i) + L_\pi^i(\Phi_\pi^i), \tag{1}$$

where we define $L_m^i$ in Eq. 9 and $L_\pi^i$ in Eq. 10.

We organize the subsections as follows. First, we introduce a Dynamic Bayesian Network in Section 4.2 to model the generation of individual rewards in POMG and provide the theoretical results of identifiability, which jointly enable us to reason the agents' contribution towards other agents' outcome. Second, we elucidate our methodology for estimating agents' counterfactual regret through counterfactual reasoning, which are integrally coupled with our innovative intrinsic reward design paradigm, as comprehensively delineated in Section 4.3. We provide the pseudo-code of our method in Algorithm 1.

## 4.2 Causal Modeling

Causality is usually exploited to model the generative process of variables in diverse systems. In this subsection, we utilize a causal model to describe the transition, observation and reward functions in the multi-agent system. The theorical identifiability supports the reliable estimation of unknown functions in the causal model given the observed data. This is especially valuable in multi-agent systems, where the complexity arises from the interactions among diverse agents and their collective impact on the system's dynamics and outcomes. Above serves as a sandstone for our counterfactual reasoning of individual rewards while agent $i$ performs the counterfactual actions.

**Generative Process in POMG.** To denote the generative process within the POMG environments, we introduce a Dynamic Bayesian Network (DBN) $\mathcal{G}$ over a finite number of random variables $[\boldsymbol{s}_t, \boldsymbol{a}_t, \boldsymbol{o}_t, \boldsymbol{r}_t] \mid_{t=1}^{T} = [\boldsymbol{s}_t, [\boldsymbol{o}_t^i, \boldsymbol{a}_t^i, r_t^i]_{i=1}^N]_{t=1}^T$, where $\boldsymbol{s}_t$ represents the latent environment state, $\boldsymbol{o}_t^i$, $\boldsymbol{a}_t^i$ and $\boldsymbol{r}_t^i$ represents the observation, action and reward of an individual agent $i$ at time step $t$. The generation process of the agent's team reward is as follows:

$$
\begin{cases}
\boldsymbol{s}_{t+1} = f(\boldsymbol{s}_t, \boldsymbol{a}_t, \epsilon_{\boldsymbol{s},t}) & \text{(environment transition function)} \\
\boldsymbol{r}_t^i = g^i(\boldsymbol{s}_t, \boldsymbol{a}_t, \epsilon_{r,i,t}) & \text{(individual reward function)} \\
\boldsymbol{o}_t^i = h^i(\boldsymbol{s}_t, \boldsymbol{a}_t, \epsilon_{o,i,t}) & \text{(individual observation function)}
\end{cases}
\tag{2}
$$

where $f$ captures the transition of environmental state; $g^i$ and $h^i$ denote the generation of $i$-th agent' individual reward $\boldsymbol{r}_t^i$ and observation $\boldsymbol{o}_t^i$, respectively. $\epsilon_{\boldsymbol{s},t}$, $\epsilon_{r,i,t}$ and $\epsilon_{o,i,t}$ denotes i.i.d random noise. Without losing generality, we assume that $\mathcal{G}$ is time-invariant, which means $f$, $g^i$, $h^i$ are time-invariant, and there are no unobserved confounders and instantaneous causal effects in $\mathcal{G}$ (Huang et al., 2021). According to the definition of POMG, $\boldsymbol{o}_t$, $\boldsymbol{a}_t$, and $\boldsymbol{r}_t$ are observable, while $\boldsymbol{s}_t$ are unobservable.

**Proposition 1.** *Suppose the observation $\boldsymbol{o}_t^i$, joint action $\boldsymbol{a}_t$, joint reward $\boldsymbol{r}_t$ are observable while latent environment state $\boldsymbol{s}_t$ is unobservable, and they form a POMG, as described in Eq. 2. Under the global Markov condition and faithfulness assumption, we can identify the causal parents of individual reward $r_t^i$, and the individual reward function $g^i$ for each agent $i$.*

**Remark 1.** *Proposition 1 establishes the theoretical identifiability of the unknown functions $f$, $g^i$, and $h^i$ in $\mathcal{G}$, based on the observed variables: observations $o_t$, joint actions $a_t$, and rewards $r_t$. This allows us to estimate the unique reward inference mapping $\Phi_m : (\boldsymbol{o}_t, \boldsymbol{a}_t) \to \boldsymbol{s}_t \to \boldsymbol{r}_t$ from the observed data, where $\boldsymbol{s}$ is reward-relevant state components, given the joint observations $\boldsymbol{o}_t$ and joint actions $\boldsymbol{a}_t$ as inputs. The proof can be found in Appendix B.*

## 4.3 Counterfactual Regret Generation

Reward shaping is widely-used technique to modify the goal of policy learning by adding an intrinsic reward term. To align with our motivation of minimizing the counterfactual regret for each agent, we present the process for performing counterfactual reasoning on other agents' outcomes, along with the computation of counterfactual regret and the design of intrinsic rewards.

**Counterfactual individual rewards.** First of all, in order to estimate the agents' counterfactual regret, the question we want to tackle is: How much would other agents earn if the agent $i$ takes the counterfactual action $\boldsymbol{a}_t^{i,\text{cf}}$, instead of $\boldsymbol{a}_t^i$? In SSD, the generation of individual rewards are impacted by all the agent's actions and the environment states. Therefore, we utilize the causal model $\Phi_m^i$ to perform the counterfactual estimation of individual rewards $\boldsymbol{r}_t^{i,\text{cf}}$ based on environment state $\boldsymbol{s}_t$ and joint action $\boldsymbol{a}_t$. The counterfactual prediction of individual rewards in the situation that agent $i$ take counterfactual actions $\boldsymbol{a}_t^{\text{cf}}$ and other agents keep their actions at state $\boldsymbol{s}_t$ can be denoted as,

$$
P(\boldsymbol{r}_t^{i,\text{cf}} \mid \boldsymbol{s}_t, do(a_t^i = a_t^{i,\text{cf}}), \boldsymbol{a}_t^{-i}) = P(\boldsymbol{r}_t^{i,\text{cf}} \mid \boldsymbol{s}_t, a_t^{i,\text{cf}}, \boldsymbol{a}_t^{-i}),
\tag{3}
$$

where $\boldsymbol{a}_t^{-i}$ denotes the actions executed by agents excluding agent $i$.

As there is no incoming causal path to action $\boldsymbol{a}_t^i$ in the causal graph and we can identify an unique mapping from the observations to the reward-relevant state components, we can estimate the counterfactual individual rewards for all agents based on joint observation $\boldsymbol{o}_t$,

$$
\boldsymbol{r}_t^{i,\text{cf}} = \Phi_m^i(\boldsymbol{o}_t, a_t^{i,\text{cf}}, \boldsymbol{a}_t^{-i}),
\tag{4}
$$

where $\boldsymbol{r}_t^{i,\text{cf}}$ is the predication of all other agents' individual rewards(For brevity, we use other agents to denote agents excluding agent $i$). Therefore, we denote $r_{j,t}^{i,\text{cf}}$ as the $j$-th element of $\boldsymbol{r}_t^{i,\text{cf}}$, which represents agent $j$'s counterfactual individual reward while agent $i$ takes counterfactual action $a_t^{i,\text{cf}}$.

Therefore, other agents would obtain a collective counterfactual reward, which is

$$R_t^{-i,\text{cf}} = \sum_{j \neq i}^{N} r_{j,t}^{i,\text{cf}} = \sum_{j \neq i}^{N} \Phi_{\text{m},j}^i(\boldsymbol{o}_t, a_t^{i,\text{cf}}, \boldsymbol{a}_t^{-i}), \tag{5}$$

where $\Phi_{\text{m},j}^i$ denotes the $j$-th element of the output vector of $\Phi_{\text{m}}$. At time step $t$, the actual collective reward other agents obtain is defined as $R_t^{-i} = \sum_{j \neq i}^{N} \Phi_{\text{m},j}^i(\boldsymbol{o}_t, a_t^i, \boldsymbol{a}_t^{-i})$.

**Counterfactual Regret.** Building upon the counterfactual reasoning of individual rewards, we could construct the counterfactual regret, $\text{Regret}_t^i$, for agent $i$ as,

$$\text{Regret}_t^i = \max_{a_t^{i,\text{cf}}} [R_t^{-i,\text{cf}}(\boldsymbol{o}_t, a_t^{i,\text{cf}}, a_t^{-i})] - R_t^{-i}(\boldsymbol{o}_t, a_t^i, a_t^{-i}), \tag{6}$$

where $a_t^{i,\text{cf}} \sim U(A)$ and $U(A)$ denotes the uniform distribution over the agent $i$'s action space. Therefore, the counterfactual regret $\text{Regret}_t^i$ measures the difference between the optimal prosocial behavior and its current behavior based on other agents' collective reward.

**Intrinsic Reward.** Recall that we want to promote the prosocial behaviors of the agents by minimizing their counterfactual regret. Therefore, we could construct the intrinsic rewards for agent $i$ as,

$$r_t^{i,\text{in}} = -\text{Regret}_t^i. \tag{7}$$

Consequencely, the reward utilized for agent $i$'s policy learning is the shaped reward $\hat{r}_t^i$:

$$\hat{r}_t^i = r_t^{i,\text{ex}} + \alpha r_t^{i,\text{in}}. \tag{8}$$

where $r_t^{i,\text{ex}}$ is the selfish individual reward from the environment and $\alpha$ is a hyper-parameter that controls how much the agent care about other agents reward.

## 4.4 OVERALL OBJECTIVES

In this subsection, we introduce the learning objectives of the generative model and the policy model.

**Generative Model Estimation** We parameterize the generative model $\Phi_{\text{m}}$ as the individual reward predictor, which takes as input the joint observation $\boldsymbol{o}_t$ and joint action $\boldsymbol{a}_t$. We optimize the generative model $\Phi_{\text{m}}$ for each agent $i$ through minimizing:

$$L_m^i = \mathbb{E}_{\boldsymbol{o}_t, \boldsymbol{a}_t, \boldsymbol{r}_t \sim D} \left[ ||\Phi_{\text{m}}^i(\boldsymbol{o}_t, \boldsymbol{a}_t) - \boldsymbol{r}_t^{\text{ex}}||^2 \right]. \tag{9}$$

**Policy learning** The shaped reward $\hat{r}_t$ enables us to train the agents' policies independently. Using PPO (Schulman et al., 2015) as the RL backbone, we minimize the following loss for each individual agent $i$. Note that the $\hat{A}$ in here is the estimated advantage function:

$$L_\pi^i = \hat{\mathbb{E}}_t \left[ \min(\hat{r}_t^i(\theta)\hat{A}_t^i, \text{clip}(\hat{r}_t^i(\theta), 1 - \epsilon, 1 + \epsilon)\hat{A}_t^i) \right], \text{ where } \hat{A}_t^i = Q_t^i(o_t^i, a_t^i) - V_t^i(o_t^i). \tag{10}$$

**Algorithm details** In our algorithm, each agent $i$ conducts their own individual policy $\pi^i$ simultaneously and gains new observations $\boldsymbol{o}_{t+1}$. We collect the observation-action pairs $\{\boldsymbol{o}_t, \boldsymbol{a}_t, \boldsymbol{r}_t\}$ for each agent at each time step $t$. After an episode ends, the model's parameters $\Phi_{\text{m}}^i$ and policy parameters $\Phi_\pi$ for each individual agent $i$ will be updated based on the sampled individual observation-action pairs and individual rewards $\{\boldsymbol{o}_t^i, \boldsymbol{a}_t^i, \boldsymbol{r}_t^i\}$.

## 5 EXPERIMENT

We conduct experiments over four SSD scenarios to demonstrate the effectiveness of our method against several baselines.

---

**Algorithm 1** Multi-Agent Counterfactual Regret Model/Policy Learning

---

**Input**: Game environment, Buffer $D = \varnothing$
**Output**: Policy set $\Pi_T$ for each individual agent, Model set $\Phi_m$ for each individual agent

1: **for** Episode $k = 0, 1, 2 \ldots K$ **do**
2:    **for** $t = 0, 1, 2 \ldots T$ **do**
3:       For agent $i = 1 : N$, conduct action $a_t^i$.
4:       Observe new observation $o_{t+1}$, reward $r_t$
5:       Add observation-action pair and individual rewards $\{o_t, a_t, r_t\}$ into buffer $D$.
6:       Predict counterfactual reward $r_t^{i,\text{cf}}$ for agents $i \in N$ using $\Phi_m^i$ based on Eq. 4.
7:       Generate counterfactual regret $-\text{Regret}_t^i$ for agents $i \in N$ as intrinsic reward via Eq. 6
8:       Add intrinsic reward $r_t^{\text{in}}$ into buffer $D$.
9:    **end for**
10:   Update model parameters $\Phi_m$ using the state-action pairs and individual rewards $\{o, a, r\}$ sampled from buffer $D$ based on Eq. 9.
11:   Update policy parameters $\Phi_\pi$ using the state-action pairs, individual rewards $\{o, a, r\}$ and predicted intrinsic reward $r^{\text{in}}$ sampled from buffer $D$ based on Eq. 10.
12: **end for**

---

### 5.1 SETUP

**Environments.** We estimate our method in four SSD environments, *Coin* (Lerer & Peysakhovich, 2017), *Level-based foraging(LBF)* (Christianos et al., 2020), *Cleanup* (Hughes et al., 2018), and *Common_Harvest* (Perolat et al., 2017). *Coin* is a three-player version of the Coin game in Melting Pot 2.0 (Agapiou et al., 2022), which was itself a version of the game introduced in Lerer & Peysakhovich (2017). There are coins corresponding to each agent scattered randomly in the environment. Whenever an agent gets a coin, they receive a reward of $+1$, but if this is not the corresponding type(e.g, agent 1 eat type 2 coin), the corresponding agent will suffer from $-2$ penalty(agent 2 will receive $-2$ penalty). In addition to the three-player version, we also include the four-player version of *Coin* in the ablation studies. In the environment *Coin_4_Agents*, we introduce one adversarial agent. Such adversarial agent have no matching coin in the environment. The detail description will be introduced in Sec 5.3. *Level-Based Foraging (LBF* is a cooperative three-player edition of level-based foraging Christianos et al. (2020). Three agents with different levels move in the grid world to consume apples. There are apples represents different levels (e.g, $1, 2, 3$). If the agents eat the apples by themselves, they only get the original value. But if the agents cooperate and consume an apple, the total reward will be multiplied by 2 in order to award cooperative behavior. We also include the four-player version of *Level-Based Foraging*. In the environment *LBF_4_Agents*, we fix the same apples number and level to create a more competitive environment. This is to test if our agents could still maintain cooperate paradigm in such intensive environment. As for *Cleanup* and *Common_Harvest*, we use the same environments as Jaques et al. (2019). We also include the 7 agents' edition for the original *Cleanup* and *Common_Harvest* environments.

**Baselinses.** We compare our method with the following baselines: the individual PPO (Schulman et al., 2015), inequity aversion (Hughes et al., 2018), SVO (McKee et al., 2020). The detailed description of the environments and baseline algorithms are deferred to the Appendix C.1.

**Metrics.** The metrics we adopt in the following experiments are the collective reward and counterfactual regret. Higher collective reward's value indicates better performance on aligning selfish agents' incentives with the team objectives. Also, lower counterfactual regret suggests the agents are more likely to conduct altruistic behaviors. Since its current action is shows more similarity to the optimal altruistic action.

### 5.2 MAIN RESULTS

We provide the main results in the Figure 2 and Figure 3, where we could see that our method shows great performance on Coin, Level-Based Foraging, Cleanup and Common_Harvest. We use Selfish to denote the individual PPO method, Inequity to denote the inequity aversion method, SVO to denote the social value orientation method, and CF to denote our method. The reason why our method

performs better than other baseline algorithms is because our method aims to maximize other agents reward under every circumstances. This would help alleviate the bias from non-related agents while calculating the total reward. We first demonstrate our results by showing our algorithm's ability under heterogeneous agents setting. *Coin* and *Level-Based-Foraging* are two simple environments that allow heterogeneous agents to cooperate in the sequential social dilemma. We also give examples on how our agents behave in setting that includes more agents.

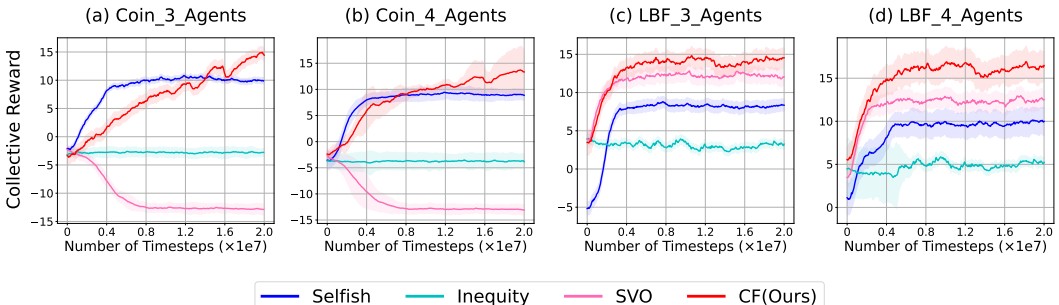

Figure 2: Learning curves on the two sequential social dilemma environments tasks(*Coin_3_Agents* and *Coin_4_Agents*) along with their variants(*LBF_3_Agents* and *LBF_4_Agents*), based on 5 independent runs with random initialization. The shaded region indicates the standard deviation. The total training steps would be $2 \times 10^7$.

In Figure 2, our Counterfactual Regret method consistently outperforms the other three baselines in all four scenarios. In the *Coin_3_Agents* and *Coin_4_Agents* scenarios, our method achieves and maintains a significant performance advantage throughout the entire training process. In *Coin_4_Agents* scenario, we aim to examine our method's ability under chaotic environments. In *Coin_4_Agents* scenario, we introduce an adversarial agent. Such adversarial agent have no matching coin in the environment. Therefore whenever it consumes a coin, it receives $+1$ reward and the corresponding agent receives $-2$ reward. Adding the adversarial agent could induce chaos into the system. We could see that though our method has more variance than 3-agents scenario, it still performs better than other three baselines. The LBF (Level-Based Foraging) scenarios further underscore the robustness of our approach. In both 3-agent and 4-agent LBF environments, our method exhibits remarkable stability and consistently higher collective rewards compared to the baselines. This performance gap is particularly pronounced in the *LBF_4_Agents* scenario, where our method maintains a substantial lead over all other methods from the early stages of training. That is because our agents tend to consider other agents' benefit more based on their counterfactual regret. Based on the counterfactual regret mechanism, our agents demonstrate better ability to cooperate with each other in more agents' setting.

In order to further demonstrate our method's ability in solving SSD, we use two classic sequential social dilemma environment(*Common_Harvest* and *Cleanup*) along with its scale up version(*Common_Harvest_7* and *Cleanup_7*). *Common_Harvest* exemplifies a classic scenario in game theory and economics known as the 'tragedy of the commons'. It involves limited common resources and several homogeneous agents aim to harvest as many resources as possible to maximize its individual reward. In Figure 3, our counterfactual regret method demonstrates exceptional performance in all four scenarios. In all cases, our method achieves and maintains the highest collective reward throughout the entire training process. This is particularly evident in the *Common_Harvest_5* scenario, where our method significantly outperforming all baselines. The *Common_Harvest_7* scenario further underscores our method's scalability, as it maintains its superior performance even with increased agent complexity. *Cleanup* is an implementation of a public goods game, where agents have to sacrifice themselves in order to achieve higher collective reward. Our method continues to demonstrate the superiority in both *Cleanup* scenarios. In both *Cleanup_5* and *Cleanup_7*, our method not only achieves the highest peak performance but also shows remarkable stability and consistent improvement throughout the training process. Notably, in the *Cleanup_7* scenario, our method demonstrates a steady upward trajectory, showing great performance in cooperative behaviors.

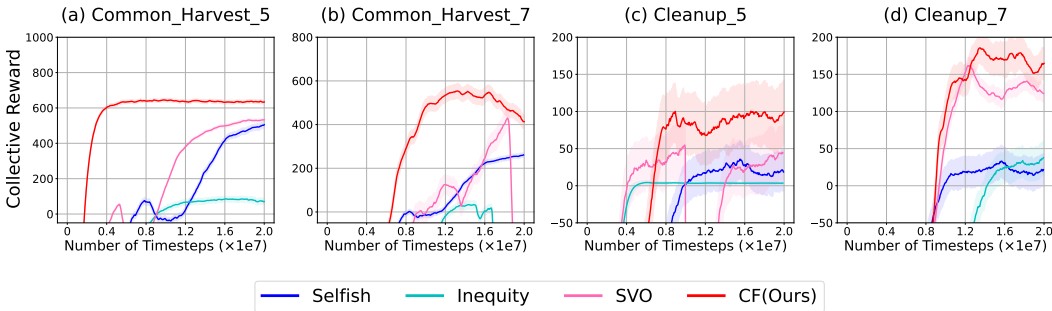

Figure 3: Learning curves on the two sequential social dilemma environments tasks(*Common_Harvest_5* and *Cleanup_5*) and their variants(*Common_Harvest_7* and *Cleanup_7*), based on 5 independent runs with random initialization. The shaded region indicates the standard deviation. The total training steps would be $2 \times 10^7$.

## 5.3 ABLATION RESULTS

In this subsection, we conduct several ablation experiments to illustrate our method's ability under different scenarios. The experiments include two parts, first, we experiment our methods under chaotic variant environments to show our method's robustness; second, we illustrate our method's ability of capturing the correct incentives for cooperative behaviors under multiple environments.

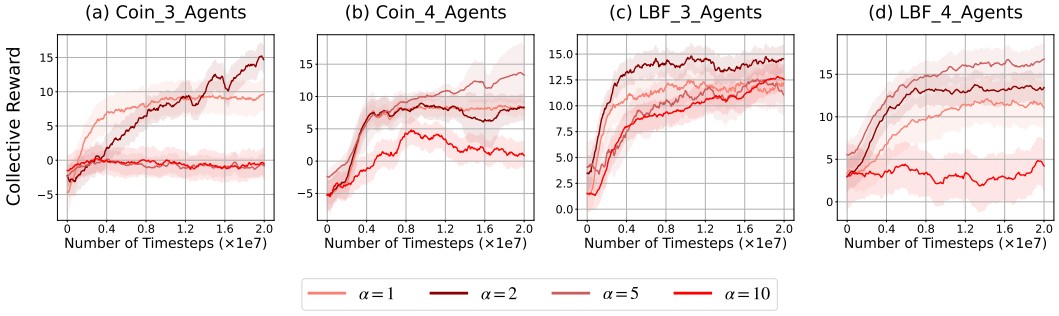

Figure 4: Learning Curves for *Coin_3_Agents*, *Coin_4_Agents*, *LBF_3_Agents* and *LBF_4_Agents* environments, with different $\alpha = \{1, 2, 5, 10\}$, based on 5 independent runs with random initialization. The shaded region indicates the standard deviation. The total training steps would be $2 \times 10^7$.

In Figure 4, we aim to measure the ability of our method under different hyperparameters $\alpha$. We could see that in the four scenarios, the optimal $\alpha$ are 2 for 3 agents setting and 5 for 4 agents setting respectively. The optimal alpha number indicates to what extent agents should care about other agents' reward. The optimal alpha suggests that each agent achieves optimal performance when it considers the collective reward of all other agents equally to its own. This can be interpreted as a form of 'fair' cooperation where an agent values the group's performance (excluding itself) as much as its individual performance. Also, this demonstrates a linear scaling of optimal cooperative behavior with the number of agents. As the system grows more complex with additional agents, the importance of considering others' rewards increases proportionally.

We aim to evaluate our method's capability to capture the correct counterfactual regret and minimize such counterfactual regret under varying hyperparameters of $\alpha$. The counterfactual regret of the agents is defined as the difference between the optimal prosocial behaviors and current behaviors. For example, in *Coin* environment, when the agents collecting their corresponding type of coin, it would be seen as a prosocial behavior, which is a behavior that is beneficial to the whole group. Therefore, as the counterfactual regret approaching 0, the agents' are more likely to conduct behaviors that are beneficial to the whole group. Therefore, in order to illustrate the method's efficacy in generating appropriate

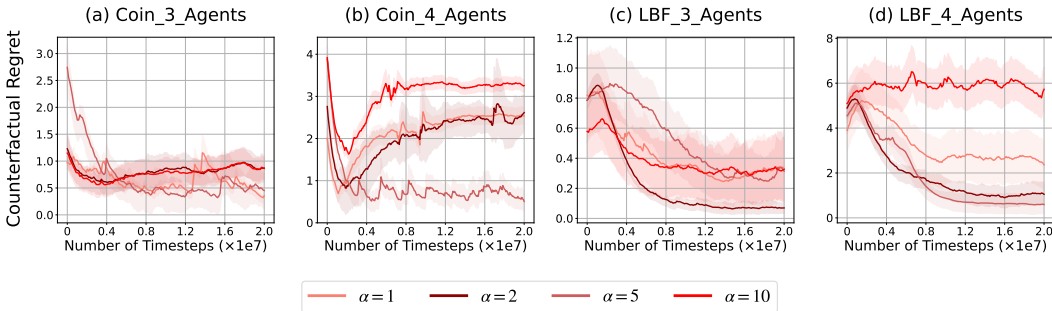

Figure 5: Illustration of counterfactual regret for *Coin_3_Agents*, *Coin_4_Agents*, *LBF_3_Agents* and *LBF_4_Agents* environments, with different $\alpha = \{1, 2, 5, 10\}$, based on 5 independent runs with random initialization. The shaded region indicates the standard deviation.

counterfactual regret, we utilize four scenarios(*Coin_3_Agents*, *Coin_4_Agents*, *LBF_3_Agents*, *LBF_4_Agents*). We set $\alpha$ to $\{1, 2, 5, 10\}$. As depicted in Figure 5, agents showed similar trend as Figure 4. Agents with higher collective reward tend to have lowest counterfactual regret. The agents' ability to generate correct incentives peaks when the hyperparameter $\alpha$ is set to $2$ in the $3$ agents setting and $5$ in $4$ agents setting. Additionally, it is observed that in simpler games(*Coin_3_Agents* and *LBF_3_Agents*), the performance for each hyperparameters $\alpha$ are similar. However, as the complexity of the game environment increases (in *Coin_4_Agents* and *Coin_5_Agents*), the choice of hyperparameters becomes increasingly critical, as indicated by the widening gap between the performance lines.

## 6 CONCLUSION

In this paper, we propose a multi-agent reinforcement learning algorithm for addressing social dilemmas by aligning agents' self-interests with the interests of others. Our approach encourages individual agents to minimize their counterfactual regret, estimated by calculating the difference between each agent's optimal prosocial behaviors and their current behaviors. This method enables agents to strike a balance between self-interest and cooperative behavior, effectively disentangling selfish rewards from prosocial ones. Empirical evaluations show that our approach consistently outperforms baseline methods in various complex social dilemma environments, demonstrating its ability to foster cooperation even in the presence of misaligned incentives and environmental complexity.

**Limitations and Future Work**   While our method has shown a promising ability to guide agents toward altruistic behavior to maximize social rewards, it presents certain vulnerabilities. Particularly when interfacing with external agents that may not share the same cooperative motives. Specifically, the altruistic nature of our agents can lead to exploitation by defectors, potentially undermining the effectiveness of our approach in competitive or mixed-motive environments. To address this critical limitation, our future work will focus on developing more robust strategies that not only promote cooperation among agents with aligned interests but also safeguard against potential exploitation.

**Ethic Statement**   In this study, we have rigorously adhered to the ICLR Code of Ethics, carefully addressing potential ethical concerns throughout our research process. Our ethical considerations encompassed three key areas: impact on human subjects, data privacy protection, and fairness in algorithmic decision-making. Robust security measures were implemented to safeguard personal information and prevent unauthorized access. We remain committed to transparency and open dialogue regarding any limitations or ethical considerations arising from our work, inviting peer review to further strengthen the ethical foundation of our research and its broader implications.

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

## A    BROADER IMPACT

The research presented in this paper tackles complex social dilemmas by developing a multi-agent reinforcement learning algorithm that aligns individual agent preferences with counterfactual collective rewards. This innovative approach represents a significant advancement in the fields of artificial intelligence and multi-agent systems. By ensuring that agents' actions are optimized not just for individual gains but for the collective good, our method has the potential to revolutionize various sectors. Societally, it can enhance cooperative behavior in automated systems, leading to more harmonious human-machine interactions. Economically, it can optimize resource allocation and decision-making processes in markets and organizations. In education, this algorithm can be used to foster collaborative learning environments and enhance adaptive learning systems. Environmentally, it holds promise for improving strategies in sustainability efforts, such as resource management and conservation initiatives. Overall, our research not only contributes to the theoretical foundations of AI but also offers practical solutions with far-reaching implications across multiple domains.

## B    DETAILS ON PROOFS

Given the joint observations $o_t^i$, $\forall i \in [1, \ldots N]$, joint action $a_t$, we prove that, reward-relevant $s_t^i$ is identifiable, as well as the unknown functions. Our generative model in Eq. 2, denoted by a Dynamic Bayesian Network (DBN) $\mathcal{G}$, is constructed over the variables $\{o_t^i, s_t^i, a_t^i, r_t^i\}^{N,T}$ in Partially Observable Markov Game.

*Proof.* According to (Pearl, 2010), we can do counterfactual reasoning if we know all the causal parents of the variable $r_t^i$. Therefore, the goal is to show that we can identify an agent $i$'s reward-relevant set of state components $s_t^{r^i}$ which have a direct path to the individual rewards $r_t^i$.

Below we show that the components $s_t^i$ in $s_t^{r^i}$ has a direct path to $r_t^i$ if and only if $s_t^j \not\perp\!\!\!\perp r_t^i \mid a_t, s_t^{\hat{r}^i}$, where $s_t^{\hat{r}^i} := \{s_t^j, \forall s_t^j \notin s_t^{r^i}\}$:

We prove it by contradiction. Suppose that $s_t^j$ is independent of $r^i$ given $a_t$, $s_t^{\hat{r}^i}$ and $R_t$. Then according to the faithfulness assumption, we can see that $s_t^j$ does not have a directed path to $r_t^i$, which contradicts the assumption, because, otherwise, $a_t$ and $s^{\hat{r}^i}$ cannot break the paths between $s_{i,t}$ and $r_t^i$ which leads to the dependence.

$\square$

**Remark 2.** *According to the proof, we can identify the individual-reward-relevant state components from the observed data, i.e., we can extract such components from the observation and learn a mapping from the observation to the individual rewards.*

## C  ADDITIONAL DETAILS ON EXPERIMENTS

### C.1  EXPERIMENT DESCRIPTION

**Level-based Foraging:**

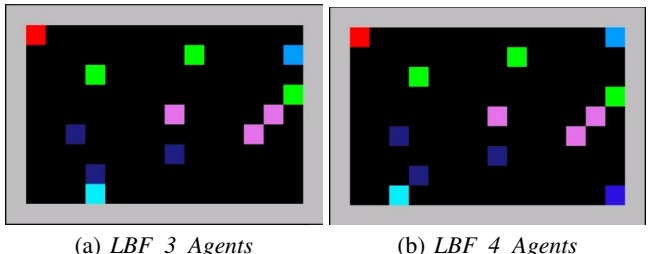

(a) *LBF_3_Agents*          (b) *LBF_4_Agents*

Agents are placed in the grid world, and each is assigned a random level. Food positions are determined in each episode, each having a level on its own(no more than 3). Agents can navigate the environment and can attempt to collect food placed next to them. The collection of food is successful only if the sum of the levels of the agents involved in loading is equal to or higher than the level of the food. Finally, agents are awarded points equal to the level of the food they helped collect(two times if they are cooperating), divided by their contribution (their level).

**Coin:**

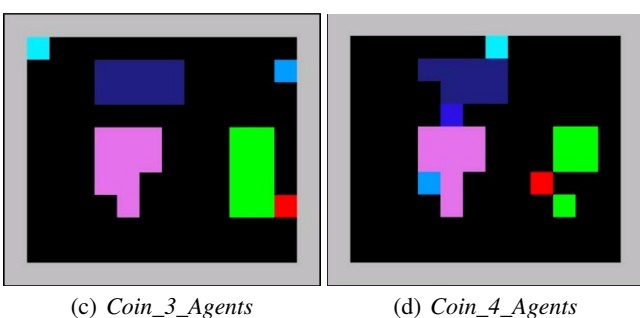

(c) *Coin_3_Agents*          (d) *Coin_4_Agents*

The reward for an individual agent in the environment at each time step under every scenario:

1. -4: other two agents get current agent's coin, while this agent does not get coin
2. -3: other two agents get current agent's coin, this agent gets a coin
3. -2: another agent get current agent's coin, this agent does not get coin
4. -1: another agent get current agent's coin, this agent gets a coin

5. 0: this agent do not get coin, other agents' do not get its coin

6. 1: this agent gets a coin

when the environment only contains two coins or one coin, the reward position of the missing reward would be (0,0), the type would also be (0). Let $C_i$ be the coin type of agent $i$. $r_i(t)$ equals to the instantaneous reward of agent $i$ at time step $t$. $S(t)$ equals to the set of all coin types in time step $t$.

$$\delta_{C_i,T} = \left\{ \begin{array}{ll} 1 & C_i = T \\ 0 & otherwise \end{array} \right.$$

Therefore, the instantaneous reward of the agent $i$ at time step $t$ is:

$$r_i(t) = \sum_{T \in S(t)} \left( \delta_{C_i,T} - 2 \cdot \sum_{j \neq i} \delta_{C_j,T} \right)$$

In the four-agent setting of *Coin*, we introduce an adversarial agent by giving it a disruptive role. This agent has no matching coin type in the environment. Its primary function shifts to disturbing the dynamics of the game, potentially interfering with other agents' actions. The optimal prosocial policy for this modified agent would be to remain stationary and abstain from coin consumption, effectively minimizing its disruptive impact. This alteration creates a more complex strategic landscape, forcing the other three agents to adapt their behaviors in the presence of a potential adversary. The scenario now balances individual coin-collecting goals against the challenge of navigating an environment with an unpredictable, disruptive element, providing a richer context for studying multi-agent interactions and conflict resolution strategies.

**Cleanup (Hughes et al., 2018):**

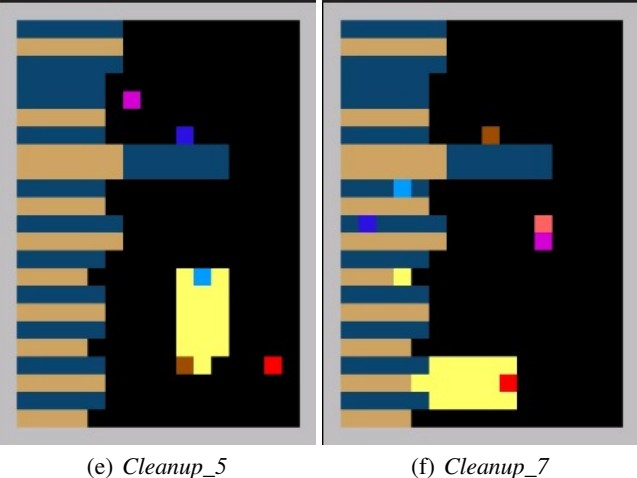

(e) *Cleanup_5*        (f) *Cleanup_7*

In *Cleanup*, all agents are equipped with a fining beam which administers $-1$ reward to the user and $-50$ reward to the individual that is being fined. There is no penalty to the user for unsuccessful fining. In Cleanup each agent is additionally equipped with a cleaning beam, which allows them to remove waste from the aquifer. Eating apples provides a reward of $1$. There are no other extrinsic rewards. In Cleanup, waste is produced uniformly in the river with probability $0.5$ on each timestep, until the river is saturated with waste, which happens when the waste covers $40\%$ of the river. For a given saturation $x$ of the river, apples spawn in the field with probability $0.125x$. Initially the river is saturated with waste, so some contribution to the public good is required for any agent to receive a reward.

We also provide the 7 agents edition for *Cleanup*. In the 7-agent edition of *Cleanup*, we expand the original environment to accommodate a larger group of participants, intensifying the complexity

of social dynamics and resource management. The core mechanics remain unchanged: agents can clean waste from the river, collect apples that spawn based on river cleanliness, and use fining beams to penalize others. However, the increased number of agents creates a more crowded and competitive space, amplifying the tension between individual and collective interests. This expanded setting challenges agents to develop more sophisticated strategies for balancing personal reward maximization with the need for cooperative cleaning efforts. The larger group size also allows for the emergence of more complex social structures, such as temporary alliances or collective punishment of free-riders. Ultimately, this 7-agent version provides a more sophisticated experimental framework for investigating how prosocial behaviors and effective resource management strategies scale in larger multi-agent systems.

**Common_Harvest (Hughes et al., 2018):**

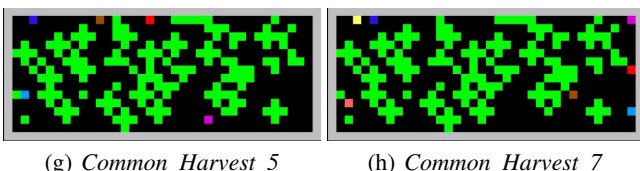

(g) *Common_Harvest_5*          (h) *Common_Harvest_7*

In *Common_Harvest*, all agents are equipped with a fining beam which administers $-1$ reward to the user and $-50$ reward to the individual that is being fined. There is no penalty to the user for unsuccessful fining. Eating apples provides a reward of 1. There are no other extrinsic rewards.

In *Common_Harvest*, apples spawn relative to the current number of other apples within an $l^1$ radius of 2. The spawn probabilities are 0, 0.005, 0.02, 0.05 for 0, 1, 2 and $\geq 3$ apples inside the radius respectively. The initial distribution of apples creates a number of more or less precariously linked regions. Sustainable policies must preferentially harvest denser regions, and avoid removing the important apples that link patches.

We also provide the 7-agents edition for the *Common_Harvest* environment. The 7-agent edition of *Common_Harvest* expands the original environment to create a more complex and challenging scenario for multi-agent cooperation and resource management. This version maintains the core mechanics of apple spawning based on local density and the use of fining beams, but introduces a larger group of agents competing for limited resources. The increased number of participants intensifies the challenge of maintaining sustainable harvesting practices, particularly in preserving the crucial links between apple patches. Agents must develop more sophisticated strategies to balance individual rewards with collective sustainability, navigating a more intricate social landscape where fining decisions and harvesting behaviors have broader implications. This expanded setting provides a richer platform for studying how sustainable resource management strategies scale with group size, the emergence of implicit social norms, and the potential for diverse role specialization among agents. Ultimately, the *Common_Harvest_7* offers deeper insights into complex multi-agent dynamics in shared resource scenarios, mirroring real-world challenges in environmental and economic systems.

In the *Common_Harvest* and *Cleanup*, agents use partially observed graphics observation, which contains a grid of $15 \times 15$ centered on themselves. Therefore, we could construct the environment as the POMG.

### C.2 ALGORITHM DETAILS

We utilized PPO algorithm in stable-baselines3 (Hill et al., 2018) to implement the baselines and our methods, with all the agents using separated policy parameters for every experiments. For SVO, we modify the individual reward to be $r_i - \alpha(1 - \arctan\left(\frac{\sum_{j,j \neq i} r_j}{r_i}\right)$ like in the original paper (McKee et al., 2020) .

The hyper-parameters for PPO training are as follows.

- The learning rate is 1e-4
- The PPO clipping factor is 0.2.

- The value loss coefficient is 1.
- The entropy coefficient is 0.001.
- The $\gamma$ is 0.99.
- The total environment step is 1e7
- The environment episode length is 1000.
- The grad clip is 40.

## C.3 COMPUTATIONAL RESOURCES

All experiments were conducted on an HPC system equipped with 128 Intel Xeon processors operating at a clock speed of 2.2 GHz and 40 gigabytes of memory.

