# OpenReview forum: "Resolving Complex Social Dilemmas by Aligning Preferences with Counterfactual Regret"
_ICLR.cc/2025/Conference — Submitted to ICLR 2025_

### Official Review · Reviewer_WiTW · 2024-10-27

**Soundness:** 2
**Presentation:** 2
**Contribution:** 2
**Rating:** 3
**Confidence:** 4

**Summary:**

The paper proposes an incentivization method for cooperation in sequential social dilemmas (SSDs) using counterfactual reasoning about the rewards of other agents. A generative model is learned to capture the reward dynamics to calculate a counterfactual regret as an intrinsic reward for prosocial learning. The proposed approach is evaluated in a variety of benchmark domains, such as Coin, Cleanup, Level-Based Foraging, and Harvest, and compared with a selection of prior methods.

**Strengths:**

The paper addresses a relevant and popular topic regarding cooperation incentivization in social dilemmas.

It is well-written and easy to understand.

**Weaknesses:**

**Soundness**
- The paper explicitly assumes a partially observable Markov game. However, given the proposed method, there is no true partial observability: Joint observations, actions, and even rewards are observable to all agents, so there is practically no privacy and all agents can fully observe each other. While they are required for the generative model, I am uncertain about potential applications where such an assumption (everything is observable) would hold.
- The observations are assumed to be Markovian since the policies condition on them directly (prior literature on POMDPs or Dec-POMDPs always consider the history of past actions and observations to mitigate this)
- All agents need to have the same "currency" of rewards. Otherwise, some agents with a significantly larger reward scale could skew the regret calculation.

**Experiments**
- Despite promoting causal inference as a main tool for the proposed approach, the paper does not compare with Social Influence, which is also based on causal inference (and cited in the paper). The paper also does not compare with alternative incentivization approaches like Gifting. Both Social Influence and Gifting achieve higher collective rewards in Harvest (over 800) than the performance reported in the paper. Social Influence also achieves higher collective rewards in Cleanup (at least 200) which is higher than the maximum performance reported in the paper.
- It is surprising that the Selfish baseline performs rather well in Coin despite performing poorly in prior works [1,2], while other approaches like SVO perform poorly despite being designed to incentivize cooperation.
- While the regret evaluations include alpha-values >= 1 (indicating selflessness), it would be interesting to see how the agents behaved if the alpha was set to something < 1, i.e., how selfish can the proposed approach be without compromising overall cooperation?
- All benchmark domains have alternative cooperation measures that can give more insight into the behavior of the agents, e.g., matching coin rate in Coin, peace/sustainability in Harvest, etc., which are not reported in the paper or appendix. I suggest to provide such plots in the main paper to strengthen the contribution and claims.

**Typos**
- "In SSDs, naively using individual for each agent" -> "reward" is missing
- "theorical" -> "theoretical"
- "rewards(For brevity" -> rewards (For brevity
- "Baselinses" -> "Baselines"

**Literature**

[1] Foerster et al., "Learning with Opponent-Learning Awareness", AAMAS 2018

[2] Phan et al., "Emergent Cooperation from Mutual Acknowledgment Exchange", AAMAS 2022

**Questions:**

1. What would be examples where full observability of all agents, i.e., their observations, actions, and rewards, is a realistic assumption?
2. How would the approach behave if, e.g., one agent in Coin would get a reward scaled by a constant factor (let's say 10), in contrast to other agents? What would need to be done to avoid bias toward that particular agent?

---

> ### Author Response · Authors · 2024-11-27
>
> Thank you for your constructive feedback. Below we provide point-wise response to your questions.
>
>
> **Response to Weakness 1:**
> > The paper explicitly assumes a partially observable Markov game. However, given the proposed method, there is no true partial observability: Joint observations, actions, and even rewards are observable to all agents, so there is practically no privacy and all agents can fully observe each other. While they are required for the generative model, I am uncertain about potential applications where such an assumption (everything is observable) would hold.
>
> Our method strictly adheres to the Dec-POMDP setting, ensuring flexibility in its application. During the inference phase, each agent's policy relies solely on its individual observations, fully aligning with the Dec-POMDP framework. While the training phase assumes access to joint observations, actions, and rewards to optimize the generative model, this does not impact the method's applicability to scenarios with partial observability.
>
> **Response to Weakness 2:**
> > The observations are assumed to be Markovian since the policies condition on them directly (prior literature on POMDPs or Dec-POMDPs always consider the history of past actions and observations to mitigate this)
>
> We do not assume the observations are Markovian in general. We compare RNN-based to encode agents' historical observations and MLP-based networks to directly encode current observation but do not observe huge performance gap, so that we choose MLP-based with less computational cost.
>
>
> **Response to Weakness 3:**
> > All agents need to have the same "currency" of rewards. Otherwise, some agents with a significantly larger reward scale could skew the regret calculation.
>
> We ensure that all agents operate on the same reward scale. Counterfactual regret captures the difference between the maximum counterfactual reward achievable by other agents and their actual rewards, avoiding issues related to scale. Its calculation involves simply taking the difference between the predicted maximum total extrinsic reward and the current total extrinsic reward. Beyond counterfactual regret, all agents share the same scale for extrinsic rewards, maintaining consistency throughout the framework.
>
> **Response to Weaknesses 4:**
> > Despite promoting causal inference as a main tool for the proposed approach, the paper does not compare with Social Influence,
>
>
> Our method is fundamentally different from Social Influence. Our primary contribution lies in the construction of counterfactual regret to foster cooperative behaviors in SSD, whereas the Social Influence approach relies on the similarity between agents' actions as an intrinsic reward. In practice, the Social Influence method employs A3C as its policy module and a CNN for perception. In contrast, we utilize PPO and CBAM, which are more efficient and capable of capturing richer information than the Social Influence framework.
>
> Regarding the test environments, we could not locate the official implementations for either method. Consequently, we relied on an unofficial version, which may account for the observed performance differences.
>
>
> **Response to Weakness 5:**
> > It is surprising that the Selfish baseline performs rather well in Coin despite performing poorly in prior works [1,2], while other approaches like SVO perform poorly despite being designed to incentivize cooperation.
>
> This arises from the rapidly changing distribution of social values, which prevents the SVO method from converging to a stable cooperative paradigm. However, if all agents rely solely on their individual selfish rewards, they will ultimately converge to a locally optimal equilibrium.
>
> **Response to Weaknesses 6:**
> > How selfish can the proposed approach be without compromising overall cooperation?
>
> In our paper, we introduce the parameter $\alpha$ to regulate the altruistic tendencies of the agents, effectively influencing their level of selfishness. Specifically, the shaped reward for agent $i$ is defined as  $\hat{r}_t^i = r_t^{i,\text{ex}} + \alpha r_t^{i,\text{in}}$ `where $r_t^{i,\text{ex}}$ represents the extrinsic reward and $r_t^{i,\text{in}}$ denotes the intrinsic reward. By adjusting $\alpha$, we control the balance between selfish rewards and contributions to social welfare. For instance, when $\alpha < 1$, the agent prioritizes its selfish rewards over the collective welfare.
>
>
>
> **Response to w7:**
> > All benchmark domains have alternative cooperation measures that can give more insight into the behavior of the agents.
>
>
> Thank you for your advice, we will provide these additional results in the next version.

---

### Official Review · Reviewer_6hGU · 2024-11-01

**Soundness:** 3
**Presentation:** 2
**Contribution:** 2
**Rating:** 5
**Confidence:** 4

**Summary:**

The paper focuses on sequential social dilemmas.  It designs a causal model to predict counterfactual individual rewards and uses counterfactual regret as an intrinsic reward to encourage prosocial behaviors. Experiments on several SSD scenarios show that the proposed method achieves higher team rewards than the baselines.

**Strengths:**

1. This paper focuses on the sequential social dilemmas that are of interest to the community.
2. The authors design a casual model to capture the generation process of individual rewards in SSDs.
3. This paper provides some theoretical analysis of the process of individual reward generation.

**Weaknesses:**

1. Some symbols appear suddenly without explanation, making them difficult to read. The sentence “To interpret the phrase: had collective actions…” seems odd and shows clear signs of AI-generated text. The entire second paragraph of Section 3 is quite confusing.
2. The paper mentions the causal model and generative model, but they seem to refer to the same model. What is the network structure of the causal model? Is the Dynamic Bayesian Network considered part of the causal model?
3. The baseline for comparison is somewhat outdated.
4. The figures in the ABLATION RESULTS are difficult to read. I recommend using more distinct colors to differentiate the curves.

**Questions:**

1. Could you give a more detailed analysis of how counterfactual regret promotes cooperation?
2. How about comparing it with the auto-aligning multi-agent incentives (Kwon et al., 2023) method mentioned in related work?
3. Why does SVO exhibit significant fluctuations in Common_Harvest_7 and Cleanup_5?

---

> ### Author Response · Authors · 2024-11-27
>
> We sincerely appreciate your constructive comments. Below we provide point-wise response to your questions.
>
>
>
>
> **Response to Weakness 1:**
> > Some symbols appear suddenly without explanation, making them difficult to read. The entire second paragraph of Section 3 is quite confusing.
>
> Thank you for pointing these. After proofreading, we revised Section 3 in the revised version of our paper
>
> **Response to Weaknesses 2:**
> > Are the causal model and generative model the same? What is the network structure of the causal model? Is the Dynamic Bayesian Network considered part of the causal model?
>
> Yes, the causal model and the generative model refer to the same concept in our work. We represent it using a Dynamic Bayesian Network, consistent with prior studies [1][2]. Therefore, Dynamic Bayesian Network is not the part of the causal model. In implementation, the network structure of the causal model is realized as a 4-layer MLP with the following dimensions: (input_dim, 1024), (1024, 512), (512, 128), and (128, output_dim).
>
> [1] Huang, B., Feng, F., Lu, C., Magliacane, S., & Zhang, K. AdaRL: What, Where, and How to Adapt in Transfer Reinforcement Learning. In International Conference on Learning Representations.
> [2] Huang, B., Lu, C., Leqi, L., Hernandez-Lobato, J.M., Glymour, C., Schölkopf, B. &amp; Zhang, K.. (2022). Action-Sufficient State Representation Learning for Control with Structural Constraints. Proceedings of the 39th International Conference on Machine Learning, in Proceedings of Machine Learning Research 162:9260-9279.
>
>
>
> **Response to Weaknesses 3:**
> > The baseline for comparison is somewhat outdated.
>
> Thanks for your suggestion. We would like to include more SOTA baselines in the next version.
>
>
> **Response to Weaknesses 4:**
> > The figures in the ABLATION RESULTS are difficult to read.
>
> Thank you for your advice, we have revised the figure through using different colors to represent different methods in the attatched version.
>
> **Response to Question 1:**
> > Could you give a more detailed analysis of how counterfactual regret promotes cooperation?
>
> Our motivation arises from the intuition that agents aim to maximize their own interests while minimizing harm to overall social welfare by reducing their regret over negatively impacting others. In this paper, we define counterfactual regret as the difference between other agents' actual outcomes and their hypothetical outcomes under alternate actions simulated by the agent. Specifically, we calculate the maximum expected outcome for other agents by predicting their optimal rewards across various counterfactual scenarios. From this, we subtract the actual rewards received by those agents to quantify counterfactual regret. By minimizing this counterfactual regret, individual agents are encouraged to adopt more prosocial behaviors by accounting for the potential outcomes of their actions on others. This approach promotes a cooperative paradigm, ultimately leading to improved collective outcomes.
>
>
> **Response to Question 2:**
> > How about comparing it with the auto-aligning multi-agent incentives (Kwon et al., 2023) method mentioned in related work?
>
> Thanks for your suggestion. However, the code is not open-source so that it is hard to make a fair comparison.
>
>
> **Response to Question 3:**
> > Why does SVO exhibit significant fluctuations in Common_Harvest_7 and Cleanup_5?
>
> The significant fluctuations of SVO in *Common_Harvest_7* and *Cleanup_5* stem from the inherent characteristics of the SSD environment. First, the interactions between agents in the SSD environment are highly fragile, leading to frequent changes in agent behavior and, consequently, rapid variations in rewards. Second, the SVO method primarily focuses on aligning each agent's preferences with the group outcome by minimizing the discrepancy between the target and current social value distributions. However, as the social value distribution in SSD environments changes rapidly, it becomes challenging for agents to achieve alignment, resulting in significant fluctuations.
>
> In contrast, our method emphasizes the individual counterfactual regret between each agent and the others. By addressing counterfactual regret on an individual basis, our approach enables agents to maintain more stable performance compared to SVO, even in dynamic SSD environments.

---

### Official Review · Reviewer_NLcZ · 2024-11-04

**Soundness:** 2
**Presentation:** 3
**Contribution:** 3
**Rating:** 5
**Confidence:** 4

**Summary:**

The paper focuses on the challenge of aligning individual desires with group objectives in situations known as Sequential Social Dilemmas (SSDs). Current research efforts to promote cooperation in SSDs are discussed, highlighting approaches that model agent interactions or incentivize altruistic behavior.

The authors propose a reinforcement learning algorithm that leverages counterfactual regret and a causal model to better align individual incentives with group goals. This approach aims to minimize biases in reward estimation by understanding the true causes of individual rewards and considering the impact of each agent's actions on others. The key contributions of this work include the development of a generative causal model for reward processes and the introduction of counterfactual regret to enhance cooperation.

**Strengths:**

1. The paper effectively highlights the limitations of previous methods in capturing the true causal relationships between agents' actions and their outcomes. By recognizing that earlier approaches often result in ineffective cooperation strategies due to delayed or spurious correlations, the authors provide a clear rationale for their research. It is reasonable to study the entanglement of agents' policies and the resulting biases in reward estimation.

The reviewer finds this argument convincing as the reviewer believes that the difficulty of SSDs lie in that the rewards are delayed and the causes of these rewards are difficult to analyses.

2. The introduction of counterfactual regret as a mechanism to align individual incentives with group objectives is a, as far as the reviewer is concerned, interesting contribution. By calculating the difference between the maximum counterfactual rewards and the actual rewards of other agents, the algorithm encourages agents to consider the broader impact of their actions.


3. The paper is well-organized, with a logical flow that makes complex concepts accessible.

**Weaknesses:**

1. By employing a causal model to guide counterfactual reasoning, the proposed method target at ensuring that counterfactual rewards are grounded in realistic and causally valid scenarios. This approach aims to minimize the risk of learning spurious relationships, thereby fostering genuine cooperative behavior among agents.

However, the proposed framework appears to lack comprehensive theoretical support. Proposition 1 does not fully—and understandably, given its complexity—rigorously substantiate the entire workflow of the proposed method.

2. As the paper is predominantly empirical, the authors should consider explicitly presenting the causal structures learned by their proposed method. Providing a clear depiction of these structures would strengthen the empirical findings and offer deeper insights into how the model operates.

3. The selection of baseline methods is currently insufficient. As the introduction references numerous related works, the paper would benefit from additional experiments to more effectively support the authors' arguments.

Incorporating a broader range of baselines would provide a more comprehensive evaluation of the proposed method's performance.

**Questions:**

Sea the previous section.

---

> ### Author Response · Authors · 2024-11-27
>
> Thank you very much for your constructive comments. We provide point-wise responses to your concerns as below.
>
>
> **Response to Weaknesses 1:**
> > The proposed framework appears to lack comprehensive theoretical support. Proposition 1 does not fully—and understandably, given its complexity—rigorously substantiate the entire workflow of the proposed method.
>
> Thank you for your question. Proposition 1 aims to fully support the entire workflow, potentially causing some confusion. To clarify: Proposition 1 establishes the identifiability of the causal parents for individual rewards using observed data. This foundational result underpins the subsequent interventions on the identified causal parents, which enable counterfactual reasoning and accurate prediction of the target variable. By bridging these steps, our theoretical results indeed support the entire pipeline of the proposed method.
>
> We revised Proposition 1 in the attached version for better understanding.
>
>
> **Response to Weaknesses 2:**
> > As the paper is predominantly empirical, the authors should consider explicitly presenting the causal structures learned by their proposed method. Providing a clear depiction of these structures would strengthen the empirical findings and offer deeper insights into how the model operates.
>
> The proposed method does not explicitly involve a causal structure. However, we acknowledge the importance of enhancing interpretability. We will include visualizations of gradient-based analyses to highlight reward-relevant state components in the future version. We believe this would strengthen the empirical findings and provide deeper insights into our method.
>
>
> **Response to Weaknesses 3:**
> >  The selection of baseline methods is currently insufficient.
>
>
> Thank you for your advice. We would like to include more baselines in the future version to demonstrate the performance of our method. However, Only the social influence paper provide an implementation for their method. None of the other baseline paper had open-sourced their code. Therefore, we will provide our version of implementation of these methods in the future version of revision paper.

---

### Comment · Area_Chair_wmKk · 2024-11-29

Dear Reviewers,

This is a friendly reminder that the last day that reviewers can post a message to the authors is Dec. 2nd (anywhere on Earth). If you have not already, please take a close look at all reviews and author responses, and comment on whether your original rating stands.

Thanks,

AC

---

### Meta-Review · Area_Chair_wmKk · 2024-12-20

**Metareview:**

The paper proposes a causal model designed to address the challenges of cooperation in Sequential Social Dilemmas (SSDs) by using counterfactual regret to align individual incentives with group objectives.

The introduction of counterfactual regret as a mechanism to align individual incentives with group objectives is interesting to the field. By encouraging agents to consider the impact of their actions on others, the method promotes cooperative behavior.

While the method introduces counterfactual regret, it lacks comprehensive theoretical support. The paper compares the proposed method to a limited set of baselines. A broader range of baseline methods would strengthen the evaluation.

**Additional Comments On Reviewer Discussion:**

The reviewers did not respond to the author's rebuttal. But, I believe the problems mentioned above are not fully addressed.

---

### Decision · Program_Chairs · 2025-01-22

Reject